# A Quick Method for Predicting Reflectance Spectra of Nanophotonic Devices via Artificial Neural Network

**DOI:** 10.3390/nano13212839

**Published:** 2023-10-26

**Authors:** Rui Wang, Baicheng Zhang, Guan Wang, Yachen Gao

**Affiliations:** Electronic Engineering College, Heilongjiang University, Harbin 150080, China; 2211795@s.hlju.edu.cn (R.W.); 2211695@s.hlju.edu.cn (B.Z.); 1213051@s.hlju.edu.cn (G.W.)

**Keywords:** nanophotonic, deep learning, reflectance spectra, neural networks, device design

## Abstract

Nanophotonics use the interaction between light and subwavelength structures to design nanophotonic devices and to show unique optical, electromagnetic, and acoustic properties that natural materials do not have. However, this usually requires considerable expertise and a lot of time-consuming electromagnetic simulations. With the continuous development of artificial intelligence, people are turning to deep learning for designing nanophotonic devices. Deep learning models can continuously fit the correlation function between the input parameters and output, using models with weights and biases that can obtain results in milliseconds to seconds. In this paper, we use finite-difference time-domain for simulations, and we obtain the reflectance spectra from 2430 different structures. Based on these reflectance spectra data, we use neural networks for training, which can quickly predict unseen structural reflectance spectra. The effectiveness of this method is verified by comparing the predicted results to the simulation results. Almost all results maintain the main trend, the MSE of 94% predictions are below 10^−3^, all are below 10^−2^, and the MAE of 97% predictions are below 2 × 10^−2^. This approach can speed up device design and optimization, and provides reference for scientific researchers.

## 1. Introduction

Nanophotonic devices refer to photonic devices with dimensions on the nanometer scale that utilize the unique properties of nanoscale structures and the light–matter interaction to control and manipulate the propagation, emission, and detection of light. These devices have been widely applied in information transmission and processing, biomedical applications, and sensing [1,2,3]. The performance and applications of nanophotonic devices are influenced by the materials, dimensions, shapes, and structures of the devices [4,5,6,7,8,9], requiring in-depth research and exploration in device design, fabrication, and characterization [10]. Traditional approaches typically involve electromagnetic simulations based on software packages [11,12], such as the finite-difference time-domain (FDTD) method or finite element modeling (FEM), to simulate the spectra, magnetic fields, and electric fields. However, the optimization of these parameters often requires manual adjustments, discarding other results, and this process can take weeks or even longer, requiring specialized expertise throughout the entire process. With the advancement of computer science [13,14,15,16,17], attention has turned to deep learning methods [18,19].

In 2015, LeCun Y et al. introduced deep learning [20], which has since experienced a tremendous breakthrough since 2012, surpassing previous techniques in image [21], speech, and text processing. Deep learning methods have also been applied in the design of nanophotonic devices [22,23,24,25,26,27,28,29,30,31,32,33,34,35]. In 2018, Peurifoy J et al. explored the application of artificial neural networks in the design of nanophotonic particles. Their network consisted of three inputs, seven outputs, and a hidden layer with five neurons, achieving accurate prediction of scattering spectra for simple nanoparticles [22]. Their network and nanoparticles are simple, but they illustrate the principle vividly. In the same year, Malkiel I et al. utilized DNN to design and characterize “H”-shaped nanostructures [24]. Their network exceeds ten layers and has a large number of nodes with big data. In fact, most neural network methods are like this. The aforementioned studies demonstrate the powerful capabilities of deep learning methods.

However, whether simple neural network structures can provide efficient prediction and classification remains a question. If simple shallow neural networks work, they will provide a faster method for people working with nonlinear data. And, due to the nonlinear relationship between nanophotonic devices and their optical responses, we use artificial neural networks to obtain the hidden relationship between the reflectance spectra and structural parameters of nanophotonic devices. This paper discusses the application of a simple three-layer shallow ANN to accurately predict the reflectance spectra trends of nanophotonic devices. Taking the Au-SiO_2_-Au structure as a simple example, model training is completed through appropriate data collection, data processing, and network training, so that structural parameters can be input into the model to quickly predict the reflectance spectra of the Au-SiO_2_-Au structure. Reflectance spectrum is a crucial indicator in the design of nanophotonic devices and is highly sensitive to the structural parameters of the devices. If a nanophotonic device has 5 parameters and 50 selectable values for each parameter (there can be billions of combinations, requiring substantial computational resources to obtain their spectra), ANN can quickly predict spectral trends using only thousands of data groups rather than traversing through millions to billions of combinations. Comparisons of model predicted values and real value results demonstrate that the model can accurately predict the reflectance spectrum trend of the Au-SiO_2_-Au structure. It is worth noting that most models are tested on a test set (a subset of the dataset) rather than on data points outside the dataset. Therefore, in this study, we conduct additional tests on structural values outside the dataset. Compared with traditional methods (numerical methods such as FDTD and FEM), this approach significantly reduces the time required for prediction. Automated prediction through models and scripts will greatly save time in collecting and organizing data and provide quick reference. It is like a mobile phone photo album: swipe through the prediction results (pictures) one by one to determine whether there is a spectral trend we need. This research extends the application of the combination of nanophotonic and deep learning, providing a rapid reference for the design of nanophotonic devices and showcasing promising prospects for future developments.

## 2. Materials and Methods

### 2.1. Nanophotonic Device Model

We adjust the dimensions of the Yao Z [36] structure and a gold ring is added to the original structure by employing the finite-difference time-domain (FDTD) method facilitated by Lumerical FDTD Solutions software [http://www.lumerical.com/tcad-products/fdtd/] (accessed on 12 May 2023), as shown in Figure 1a,b. The substrate is Si, the intermediate metal material is Au, and SiO_2_ serves as the electrolyte. All three have equal length and width, with x = y = 200 nm. The thickness of the Si substrate is H_Si_ = 150 nm, the thickness of Au on the Si layer is H_Au_ = 100 nm, and the thickness of SiO_2_ is H_SiO2_ = 30 nm. The total thickness of the three layers is H = 280 nm. There is a gold circle a_1_ and a gold ring a_2_ on the top that can be controlled separately for radius and height. For convenience of statistical analysis, we set specific structural parameter spacings. However, in fact, there are countless combinations within the range of values taken by h_2_, r_o_, r_i_, h_1_, and r_1_. Following are the parameter combinations we set.

The initial radius of the gold circle is r_1_ = 10 nm. The radius is increased by 10 nm in each of the 8 cycles, resulting in radii of 10 nm, 20 nm, …, 80 nm, and 90 nm. The initial height of the gold circle is h_1_ = 25 nm. The height is increased in 2 cycles by 25 nm each time, resulting in heights of 25 nm, 50 nm, and 75 nm for the gold circle. The initial inner radius of the gold ring is r_i_ = 90 nm. The inner radius is increased by 10 nm in each of the 4 cycles, resulting in inner radii of 90 nm, 100 nm, 110 nm, 120 nm, and 130 nm. The initial outer radius of the ring is r_o_ = 140 nm. The outer radius is increased by 10 nm in each of the 5 cycles, resulting in outer radii of 140 nm, 150 nm, 160 nm, 170 nm, 180 nm, and 190 nm. The initial height of the gold ring is h_2_ = 25 nm. The height is increased in 2 cycles by 25 nm each time, resulting in heights of 25 nm, 50 nm, and 75 nm for the gold ring. There is a total of t = 2430 combinations obtained from arranging and combining different radii and heights of the circle and ring. A plane wave light source is placed above the material and is perpendicular to the X-Y plane and parallel to the *Z*-axis, with a wavelength range of 400–700 nm. At the same time, periodic boundary conditions are set in the X and Y directions, with a FDTD mesh accuracy of 2, the minimum wavelength for calculation is 0.1 nm, and mesh accuracy is an indicator of the mesh size used by FDTD adaptive mesh. There are 8 accuracy levels (2 corresponding to 1/10 of the minimum wavelength) and 100 monitoring frequency points from 400 nm to 700 nm.

We use FDTD script to carry out batch simulations, with one parameter changed in each cycle. The order of parameter changes is h_2_, r_o_, r_i_, h_1_, and r_1_. A total of 2430 simulations are carried out, taking 45 h. During each individual simulation, the data are automatically collected using a script, with a simulation time ranging from 60 to 200 s. In each simulation, 5 geometric parameters and 100 spectra reflectance parameters are obtained, forming the data set.

We input over 2000 sets of simulation data into the neural network. Though simulations also require a considerable amount of time and computing resources, once the neural network is trained, the model can accurately predict the spectra generated by countless combinations of structures in just a few milliseconds to seconds.

### 2.2. Artificial Neural Network

Within the realm of deep learning, supervised learning stands as the prevailing approach. In this paradigm, a data set comprises multiple samples, with each sample containing an input and its corresponding output. The structure of a basic neural network encompasses an input layer, hidden layers, and an output layer, as depicted in Figure 2a. In the context of our paper, a reflectance spectrum is represented by a series of discrete points, as illustrated in Figure 2b. For the implementation of our work, we leverage the PyTorch framework [14], which provides a robust and efficient platform for our deep learning experiments.

Artificial neural network (ANN) is a computational model designed to fit the structure and function of biological neural networks. It consists of many neural units that interact with each other through weighted connections, forming a hierarchical structure. ANN can be used in various fields such as pattern recognition, classification, data mining, natural language processing, and image processing, and has become an important component of machine learning. The most basic building block of an ANN is the neuron. A neuron receives inputs from other neurons, weights these inputs by their respective connection strengths, and adds a bias value before feeding the result into an activation function for nonlinear transformation. The output from one neuron can then serve as an input to subsequent neurons, thus forming layers of interconnected neurons that constitute the neural network. In ANN, a backward propagation algorithm is commonly used for repeated iterations on a train set to adjust the connection weights and bias values, minimizing the error between network outputs and real results. After training, ANN can be used for prediction and classification of new data [18,19,37]. The advantages of ANN include good nonlinear fitting ability, adaptability, robustness, and generalization capabilities, while its disadvantages include a tendency to overfit, the need for large amounts of training data, and computation time. In recent years, with the rise of deep learning, deep neural network (DNN) has become a widely used form of ANN and has achieved many significant research outcomes [30,38].

When a neural network is more inclined to fit a particular set of data but fails to fit unseen data, it is called overfitting. Adjusting hyperparameters can improve the model’s fitting ability, preventing overfitting and underfitting. Following are some important parameters in a neural network such as loss function, epoch, learning rate, activation function, and number of nodes [39,40].

(1) The loss function is used to measure the gap between a model’s predicted output and real output. In deep learning, a loss function is typically used to guide the model’s learning process by minimizing the loss value, thereby improving the accuracy of the model. Different tasks require different types of loss functions. Cross entropy is commonly used as a loss function in classification problems, while mean squared error or mean absolute error is generally employed in regression tasks. The selection of a loss function is also influenced by the complexity of the model architecture and task requirements.

(2) Epoch refers to the process where the entire training data set has been trained once. During the training, the training data are usually divided into batches, with each batch containing multiple samples. The model performs operations such as forward propagation, calculating loss function, backward propagation, and parameter updating on each batch until all batches in the training data set have been used to complete one epoch. In deep learning, multiple epochs are often required to train models to improve accuracy and generalization.

(3) Learning rate is a hyperparameter used in neural network training to control the step size taken when updating weights during each iteration (or each epoch). A larger learning rate results in a bigger step size, accelerating model convergence, but may cause the model to oscillate around the optimal point or even fail to converge. Conversely, a small learning rate slows down model convergence, requiring more iterations to reach the optimal point. Therefore, choosing a reasonable learning rate is crucial for neural network training. Typically, the optimal learning rate is selected through experiments.

(4) Activation function transforms input signals into output signals. In neural networks, activation functions are typically used to introduce nonlinear characteristics to neurons, allowing the network to model more complex data. Popular activation functions include Sigmoid function, Tanh function, ReLU function, etc. These three activation functions can be represented using mathematical formulas and shown in Figure 3:(1)$$\mathrm{Sigmoid}:\sigma \left(x\right)=\frac{1}{1+e^{-x}}$$
(2)$$\mathrm{Tanh}:\sigma \left(x\right)=\frac{e^{x}-e^{-x}}{e^{x}+e^{-x}}$$
(3)Re⁡LU:σx=max0,x

(5) The number of nodes typically refers to the number of neurons in each layer of a neural network. In the input layer, the number of nodes is equal to the dimensionality of the input data. In the output layer, the number of nodes is equal to the dimensionality of the output results. Increasing the number of the nodes in the hidden layers theoretically enhances the model’s expressive power, but it also increases training time and resource consumption and carries the risk of overfitting.

(6) An optimizer is an algorithm used to update the model parameters with the aim of minimizing the loss function [41]. The choice of optimizer impacts the convergence speed and accuracy of the model [42]. Here are some commonly used optimizers:

Stochastic gradient descent (SGD): It updates the parameters using the gradient of a single sample at a time. It is suitable for small data sets [20].

Batch gradient descent (BGD): It updates the parameters using the gradient computed from the entire data set. It is suitable for large data sets but has a slower convergence speed.

Adam optimizer: The Adam optimizer can be considered as one of the most widely used optimization algorithms due to its fast convergence and stable convergence process. It requires only first-order gradients and consumes minimal memory.

### 2.3. Training Process

In deep learning, training sets, validation sets, and test sets are data sets used for training models, optimizing models, and evaluating the performance of a model. Typically, the training set constitutes most of the data set and is used to train the model. The validation set is usually a randomly selected portion of the training set. The test set is an independent data set that has never been exposed to the model and is used to evaluate the performance of the deep learning model.

In Section 1, we collect a data set of 2430 samples by FDTD. Each sample consists of a set of inputs, called features, and a set of reflectance spectra outputs, called labels, which form the data set. The radii and heights of the top ring and circle of the device are allocated as input variables to the nodes of the input layer, and the reflectance spectra corresponding to each set of input variables are taken as the output variables. The data collection process takes 45 h using a computer configuration of 2.90 GHz 16 G CPU and RTX3060 12 G GPU.

During the data preparation phase, we randomly split the collected 2430 samples into two separate data sets: a training set and a test set. The training set accounts for 90% of the data set, while the test set accounts for 10%.

During model training, we further split the previously defined training set into a training subset and a validation subset. The validation subset is randomly sampled from the training set, typically using a 70% training and 30% validation ratio, to which we also adhere. After training the model on the training subset, we calculate metrics such as the loss function on the validation subset to assess the model’s performance and identify areas for improvement. If the model performs poorly on the validation subset, we adjust and optimize the model hyperparameters to enhance its performance. Finally, we evaluate the model’s prediction capabilities using the test set.

### 2.4. Artificial Neural Network Structure

Our artificial neural network (ANN) is structured with 3 hidden layers, each consisting of 300 nodes, as depicted in Figure 4a. Having a smaller number of nodes helps prevent overfitting and ensures computational efficiency. We utilize ReLU as the activation function, which is a common choice for deep learning due to its ability to model complex nonlinear relationships.

To train our network, we employ the Adam optimization algorithm. Adam is well-suited for tasks like ours, offering a good balance between computational efficiency and convergence speed. The input layer of our neural network comprises 5 nodes. These nodes represent the radii and heights of a1 and a2 (r_1_, h_1_, r_i_, r_o_, h_2_). Our output layer is made up of 100 nodes, corresponding to discrete reflectance spectra points spanning from 300 nm to 700 nm. These nodes effectively capture the spectral information that is generated as a result of the input parameters. In the context of this paper, we treat the spectral prediction as a regression problem. We assess the performance of our model primarily using the MSE as the loss function. A lower MSE indicates a better fit of our model to the data.

To optimize the training process, we set the learning rate to 0.001. This rate controls the step size during training and is chosen to balance training speed and stability. Furthermore, we employ a batch size of 32, which means that we update the weights of the neural network after every batch of 32 data points, a common practice to enhance training efficiency.

## 3. Results

Mean squared error (MSE) is widely used in regression problems and represents the mean of the squared differences between the predicted values and the real values. In the context of predicting discrete spectra points, we treat it as a regression problem and use MSE to evaluate the accuracy of the prediction model. A higher MSE indicates larger prediction errors, while a lower MSE indicates smaller prediction errors. During model optimization, the model parameters are adjusted to minimize the MSE, aiming to make the model predicted values $\overset{\wedge }{yi}$ closer to the real values yi, where n is 100 that represents 100 discrete spectra values. The formula for MSE is as follows:(4)$$\mathrm{MSE}=\frac{1}{n}{\sum_{i=1}^{n}\left(\overset{\wedge }{yi}-yi\right)}^{2}$$

The loss curve of the network is shown in Figure 4b,c. The training process is completed with 6000 epochs, and the final values of the training loss and validation loss are 3.0271 × 10^−4^ and 7.1863 × 10^−4^, respectively. The neural network shows ideal training and validation results.

### 3.1. Testing of the Test Set

After obtaining the optimal hyperparameters, the final step is to evaluate the performance of the trained model on a previously unseen test set, which is generated separately from the initial data set. It has 243 groups (10% of 2430). We use the trained model to predict the reflectance spectra for different radii and heights and compare them to the FDTD results. We measure the difference between the predicted values and the real values using MSE and mean absolute error.

The mean absolute error (MAE) is a common metric used to evaluate the difference between predicted values $\overset{\wedge }{yi}$ and real values yi. It measures the average absolute difference between the predicted values $\overset{\wedge }{yi}$ and real values yi. n is 100 that represents 100 discrete spectra values. The formula for MAE is as follows:(5)$$\mathrm{MAE}=\frac{1}{n}\sum_{i=1}^{n}\left|\overset{\wedge }{yi}-yi\right|$$

Figure 5 shows the MSE and MAE between structural FDTD simulation and prediction in the test set. The vast majority of MSEs are less than 0.001, and the vast majority of MAEs are less than 0.02. In this case, we can obtain the logarithmic form of the results for both MAE and MSE by taking the logarithm (log) of their respective values:

The predicted results unequivocally showcase the proficiency of the artificial neural network (ANN) model in accurately predicting the reflectance spectra of photon devices. Figure 6 provides a comprehensive comparison between the representative ANN-predicted spectra results and the corresponding spectra obtained through FDTD simulations. Specifically, we have selected four samples to show the efficacy of the ANN model. Figure 5b shows that the MAE of 97% of the prediction results is less than 0.02. Figure 6a shows the prediction result with an MAE of 0.0365. It has a larger MAE, much larger than 97% of other predictions. And, due to the FDTD Lumerical (2018a) software, wavelength data are not in an arithmetic sequence (see the discrete point positions in Figure 2b, which are dense at the front and sparse at the back), so all prediction results appear to have larger errors in the back half.

Figure 6 illustrates some prediction results, and it can be observed that the majority of the results follow the main trends. This observation highlights the high reliability and practicality of the ANN model in capturing the essential features of the reflectance spectra. The model only takes about 33 s to obtain the discrete spectral data and comparison images for the 243 sets of structural parameters in the test set. On average, it only requires 0.13 s per set of structural parameters, significantly faster compared with the traditional methods that take 60–200 s (for a single simulation of the structure in this paper, it takes 60–200 s to use FDTD Lumerical).

### 3.2. Testing Outside the Data Set

In our paper, we utilize a data set to train and evaluate our model. However, to test the generalization capability of the model, we also conduct tests on structures that are outside the dataset, within the range of r_1_, h_1_, r_i_, r_o_, and h_2_. Figure 7 presents two predicted results that maintain the main trends: the structure in Figure 7a, where r_1_ = 50 nm, h_1_ = 49 nm, r_i_ = 131 nm, r_o_ = 172 nm, and h_2_ = 74 nm, and in Figure 7b, where r_1_ = 11 nm, h_1_ = 75 nm, r_i_ = 130 nm, r_o_ = 161 nm, and h_2_ = 49 nm.

### 3.3. Model’s Application Potential

The above paragraph deals with the relevant tests of the model’s accuracy. During this section of the testing, we also discover that, when r_1_ = 49 nm, h_1_ = 49 nm, r_i_ = 100 nm, r_o_ = 142 nm, and h_2_ = 52 nm (parameters that are not included in the previously collected dataset), and this structure exhibits significant absorption capabilities. We validate this structure using the FDTD method, as shown in Figure 8. Although this structure is presented merely as a straightforward example to illustrate the neural network approach, it is evident that this structure exhibits an excellent absorption performance between 400 and 570 nm. The absorption rate approaches 1, indicating minimal energy loss in this wavelength range and high stability, making it suitable for absorber applications.

However, it is important to note that the material’s polarization properties consistently affect the absorber’s absorption rate. Therefore, we conduct an analysis of absorption rate variations by altering the incident light’s angle. Setting the incident angle θ at 0°, 10°, 20°, and 30°, it becomes apparent that within the 400–570 nm wavelength range, when the polarization angle is 10°, the average absorption rate can be maintained at 90%. This demonstrates that the absorber possesses a strong absorption rate. As the polarization angle increases, the absorption rate gradually decreases. Hence, adjusting the polarization angle of the light source provides a means of controlling the absorption rate.

## 4. Discussion

In this paper, the main goal is to use the deep learning method, especially the neural network, to quickly and accurately predict the reflectance spectra of nanophotonic devices, taking a simple Au-SiO_2_-Au structure as an example. We design an artificial neural network, and through a series of tests, the accuracy and rapidity of the artificial neural network in predicting the reflectance spectra are verified.

To train the ANN, a small yet diverse data set is created by varying the radii and heights of the gold circle and gold ring. This data set serves as a robust foundation for training the network. A simplified ANN architecture with three hidden layers is selected, allowing for efficient training and deployment of the model.

The test results showcase impressive performance metrics. Approximately 94% of the predictions achieve an MSE below 10^−3^, while all predictions within the test set exhibit an MSE below 10^−2^ across the wavelength range of 300 nm to 700 nm; 97% of the predictions demonstrated an MAE below 2 × 10^−2^ in the same wavelength. Importantly, we also conduct tests on structures outside the dataset to test model performance. The trained neural network model preserves the main features and trends of reflectance with respect to wavelength, ensuring the fidelity of the predicted spectra.

The trained ANN model demonstrates astonishing speed, capable of obtaining intuitive spectral data for any parameters within the range in just 0.13 s, achieving a significant speed improvement compared with the traditional numerical methods that take 60–200 s. Moreover, it maintains the main spectral trends. This effective prediction method holds tremendous prospects for optimizing the design of nanophotonic devices and reducing associated costs.

## Figures and Tables

**Figure 1 nanomaterials-13-02839-f001:**
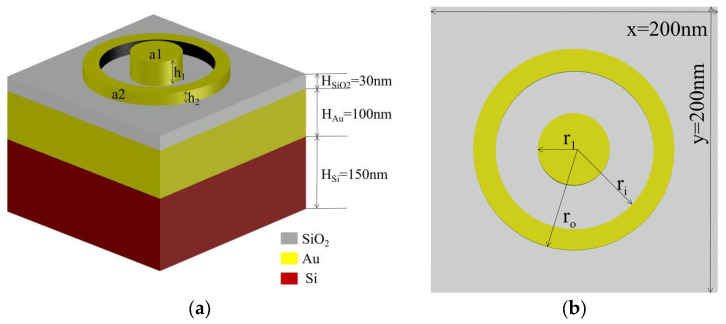
(**a**) Device 3D visualization of the structure; (**b**) top-view layout of the structure.

**Figure 2 nanomaterials-13-02839-f002:**
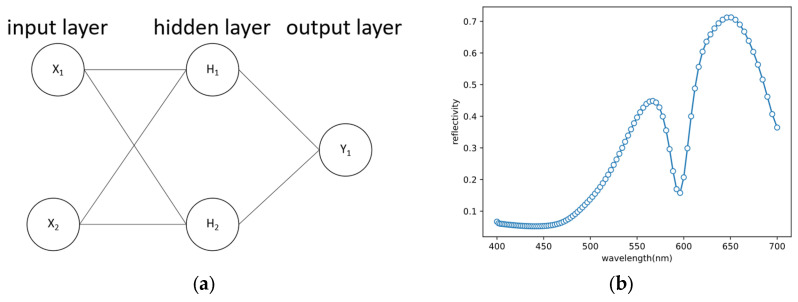
(**a**) A simple neural network. (**b**) Reflectance spectrum represented by discrete reflectance spectrum data points in this paper.

**Figure 3 nanomaterials-13-02839-f003:**
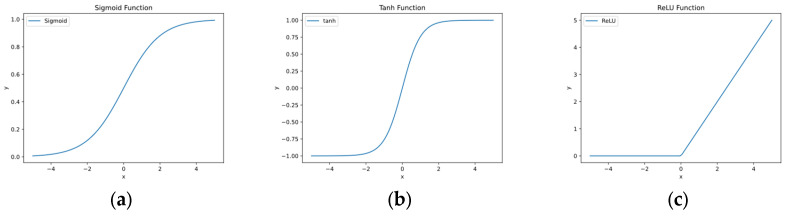
(**a**) Sigmoid function and (**b**) Tanh function are commonly used continuous, smooth, and monotonically increasing functions. (**c**) ReLU function is a nonlinear, simple, and computationally efficient function that effectively solves the vanishing gradient problem.

**Figure 4 nanomaterials-13-02839-f004:**
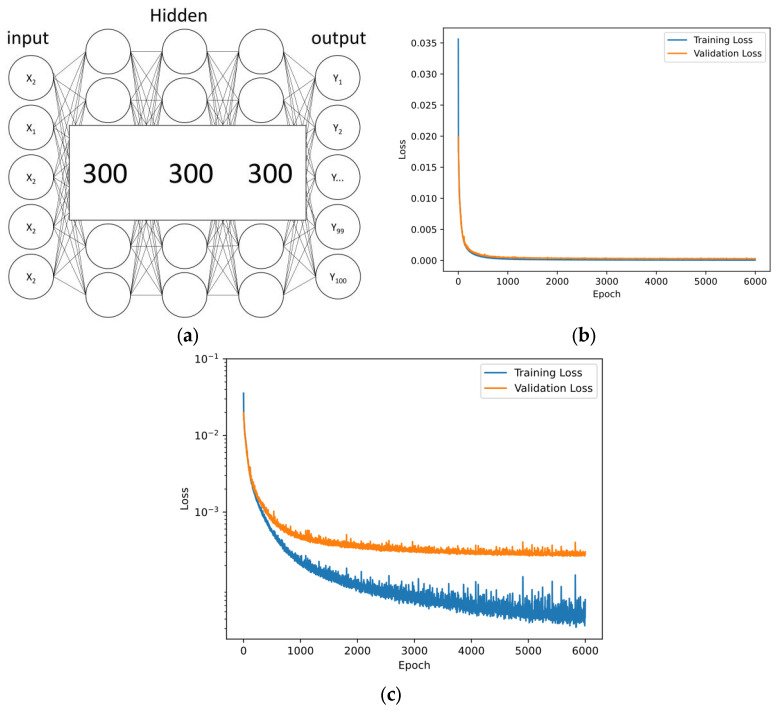
Artificial neural network (ANN) and loss. (**a**) Network structure, one input layer, three hidden layers, and one output layer. (**b**,**c**) Training and validation loss after 6000 epochs with MSE as the loss function. The final MSE values of the training loss and validation loss are 3.0271 × 10^−4^ and 7.1863 × 10^−4^.

**Figure 5 nanomaterials-13-02839-f005:**
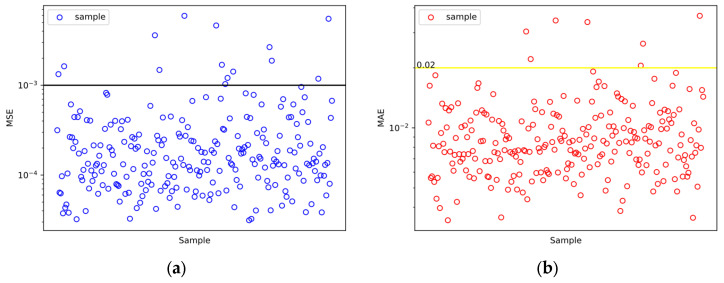
The distribution of MSE and MAE for the test set. (**a**) The MSE between the predicted and real values, all of the MSEs below 10^−2^, black line can be observed that 94% MSEs of the test set are below 10^−3^. (**b**) The MAE between the predicted and real values, yellow line represents the MAE of 97% of the test set below 2 × 10^−2^.

**Figure 6 nanomaterials-13-02839-f006:**
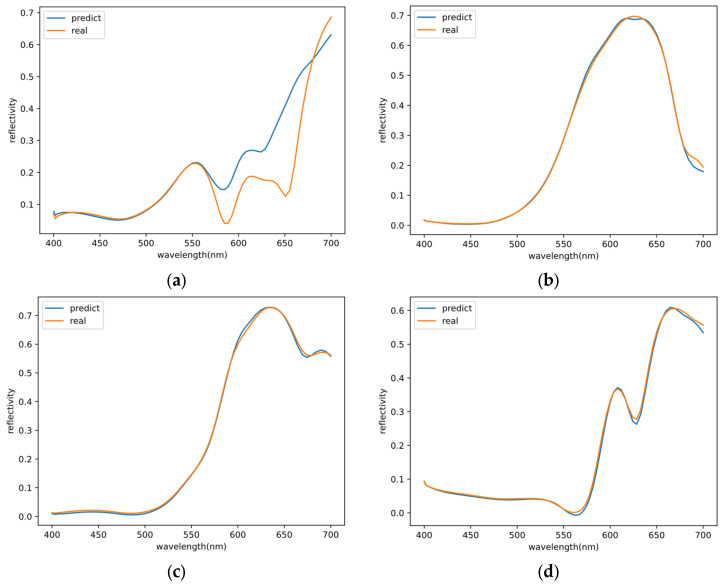
Comparison of predicted and real reflectance spectra values. (**a**) Sample MAE is 3.65 × 10^−2^, where r_1_ = 90 nm, h_1_ = 50 nm, r_i_ = 90 nm, r_o_ = 140 nm, and h_2_ = 50 nm. (**b**) Sample MAE is 3.54 × 10^−3^, where r_1_ = 20 nm, h_1_ = 25 nm, r_i_ = 130 nm, r_o_ = 170 nm, and h_2_ = 75 nm. (**c**) Sample MAE is 8.89 × 10^−3^, where r_1_ = 70 nm, h_1_ = 25 nm, r_i_ = 130 nm, r_o_ = 190 nm, and h_2_ = 75 nm. (**d**) Sample MAE is 4.41 × 10^−3^, where r_1_ = 60 nm, h_1_ = 75 nm, r_i_ = 110 nm, r_o_ = 190 nm, and h_2_ = 50 nm.

**Figure 7 nanomaterials-13-02839-f007:**
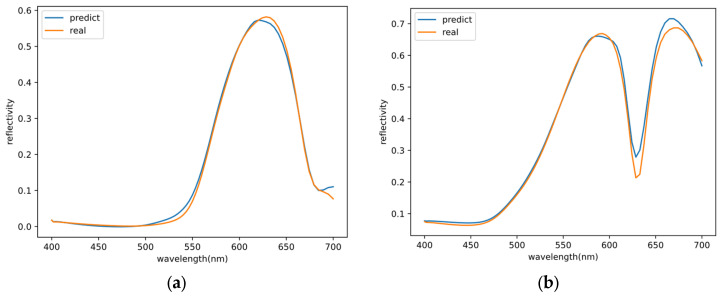
Comparison of predicted and real reflectance spectra values. Sample (**a**), where r_1_ = 50 nm, h_1_ = 49 nm, r_i_ = 131 nm, r_o_ = 172 nm, and h_2_ = 74 nm. Sample (**b**), where r_1_ = 11 nm, h_1_ = 75 nm, r_i_ = 130 nm, r_o_ = 161 nm, and h_2_ = 49 nm.

**Figure 8 nanomaterials-13-02839-f008:**
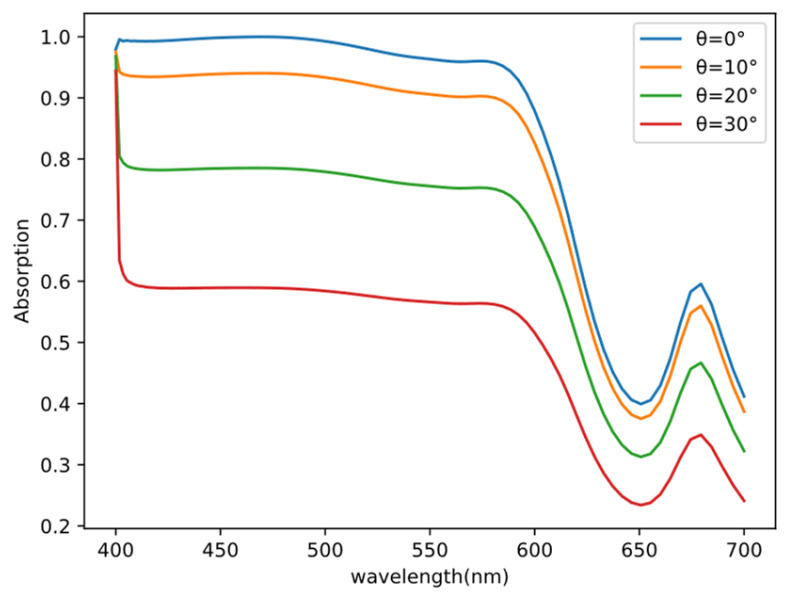
At r_1_ = 49 nm, h_1_ = 49 nm, r_i_ = 100 nm, r_o_ = 142 nm, and h_2_ = 52 nm, the absorption rates at different polarization angles (θ) of 0°, 10°, 20°, and 30° are examined. It can be observed that the absorption rate decreases with an increase in the polarization angle. Within the 400–570 nm wavelength range, at θ = 0° and 10°, this structure exhibits an absorption rate of over 90%.

## Data Availability

All content and data have been displayed in the manuscript.

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
