# Peer review of "A Quick Method for Predicting Reflectance Spectra of Nanophotonic Devices via Artificial Neural Network"

_nanomaterials, 2023, doi:10.3390/nano13212839_

Round 1
Reviewer 1 Report
Comments and Suggestions for Authors
Dear Authors,
Unfortunately, the current publication requires a significant amount of improvement before getting published. I will not be able to recommend publication at this point but I would be more than happy to reconsider after your improvements.
My main problem of your submission is about the device that you model and analyze. A deep learning algorithm should be there to offer us a guidance to obtain a nanophotonic device that for example almost nullifies the reflections from 400-700nm, which is a short range for realization relatively despite the hardship coming from gold related dispersive losses. You can find numerous such examples from the literature (eg. https://doi.org/10.1515/nanoph-2019-0474, https://doi.org/10.1515/nanoph-2021-0713, https://doi.org/10.1021/acs.nanolett.9b03971) where random looking structures are utilized/created from fundamental blocks to serve a certain nanophotonic purpose (eg. beam splitting, total absorption etc.). Compared to the structures given in various reviews in the literature in such as https://doi.org/10.1364/PRJ.415960, I cannot determine the specific use of the immediate nanophotonic device, moreover your device at hand is significantly simplistic to investigate as one would objectively point out. Are we illuminating at normal incidence? In that case, one can almost write down an analytical formulation of the diffraction theory let alone a need for a deep learning algorithm. It would be a diffraction from an annular geometry. Why would the researchers really be invested in the configuration that you studied throughout the paper? I cannot determine the impact for the field. Instead, my suggestion is to apply it to much more complicated structures. Would the algorithm be able to predict the outcome of such complicated structures as well? The studied case surely takes some time (45hours as indicated) but it is nevertheless still a manageable one with a 2-D periodicity.
Other points, please pay attention to the following items listed for clarifications:
1) We need an apples to apples comparison. Hence, we need to give concrete demonstration of the improvement with the employed algorithm. Certain references must be presented to show that "traditional methods" yielding 60-200 seconds have dropped down to 0.13 seconds. We need clear demonstration, which will underline the importance of the algorithm.
2) Last part of the introduction talks about the experimental verification. The word experiment is generally reserved for actual hands-on realization and measurement on the generated devices. Please change that to numerical realization or something similar that the authors would prefer.
3) FTDT mesh accuracy on p.3. This should be clarified as it is a Lumerical specific parameter. What does this accuracy point to?
4) Generally, the capital letter T is reserved for transmission. However, it is used for reflection in Fig. 2b.
5) Radius r1 should be clearly showed as the radius not the diameter on Fig. 1b.
6) Figure label in Figure 3(c) is missing. There is no (c).
7) Authors already mentioned ANN in earlier pages. No need to use the long form on page 4.
8) Why are we reserving a huge length of space to discuss the fundamentals of the neural networks in a very general perspective on page 4 and 5? They could have been much shorter by focusing specifically on the particulars of the current method. Many more references could have been presented if the authors would like to keep these discussions.
9) MSE formula has the square really out of the space in terms of formatting.
10) Fig. 6a is still very much further away. There is significant disagreement between the result of the ANN and the numerical result. How should I explain it?
Comments on the Quality of English Language
I did find a couple of formatting related mistakes. But in general the used English had no major issues. A careful Word based proofreading should solve those problems.
Reviewer 2 Report
Comments and Suggestions for Authors
I recommend the authors consider more recent works in the field such as [Nanophotonics 9.5 (2020): 1189-1241], [Nature communications 13.1 (2022): 1696], [Nano letters 21.3 (2021): 1238-1245] in which authors have demonstrated the significance of ANN in the design of optimized reconfigurable nanophotonic structures. This gap should be filled in the introduction section by highlighting the potential of AI in nanophotonics.
Reviewer 3 Report
Comments and Suggestions for Authors
The paper reports on a study to extract the photonic structure based on its reflectivity spectrum using ANN approach and pre-established database. The inverse problem approach is well known and have been used in the past by many including the use ANNs in variety of fields, here I mention some:
1. In optical scatterometry used for inspection of production of nanoelectronic devices, see papers:
I. Kallioniemi, J. Saarinen, and E. Oja, “Optical scatterometry of subwavelength diffraction gratings: neural-network approach,” Appl. Opt. 37, 5830–5834 (1998).
I. Abdulhalim, Simplified optical scatterometry for periodic nanoarrays in the near-quasi-static limit, Vol. 46, No. 12, APPLIED OPTICS 2219, 2007
2. In the field of computational spectral imaging see papers:
Y. LeCun, Y. Bengio, G. Hinton, Nature 2015, 521, 436.
G. Barbastathis, A. Ozcan, G. Situ, Optica 2019, 6, 921.
Doron Pasha, Marwan J. Abuleil, et.al., Lasers&Photonics Review 2200913(8p) (2023).
3. In the field of thermochromic nanomaterials design see:
Igal Balin, Valery Garmider, Long Yi and Ibrahim Abdulhalim, Training artificial neural network for optimization of nanostructured VO2 based smart window performance, Optics Express, 27(16), 1030—040 (2019).
Considering the fact that the field of AI and its uses in nanophotonics is a hot topic now, I think worth publishing the paper, however the authors should give the credit to old works as well as required.
Finally worth giving more explanation on the algorithm used.
Round 2
Reviewer 1 Report
Comments and Suggestions for Authors
Dear Authors,
Thank you for reformulating the work and most significantly carefully setting the objective as a prediction attempt by neural networks of the simple structure. This will definitely help the readers to understand the set goals and would not make an unfair comparison between your study and some of the very top level reverse engineering based applications. I also appreciate you looking into the application side of the things and adding the figure. I did understand now why you wanted to show the larger errors intentionally, too. I think at this point, the article is ready for publication.
Reviewer 3 Report
Comments and Suggestions for Authors
Following the modifications by the authors the paper can be accepted.